# Lower Limb Paralysis Associated with Chikungunya in Kinshasa, the Democratic Republic of the Congo: Survey Report

**DOI:** 10.3390/pathogens13030198

**Published:** 2024-02-23

**Authors:** Mathy Matungala-Pafubel, Junior Bulabula-Penge, Meris Matondo-Kuamfumu, Samy Esala, François Edidi-Atani, Elisabeth Pukuta-Simbu, Paul Tshiminyi-Munkamba, Yannick Tutu Tshia N’kasar, Trésor Katanga, Etienne Ndomba-Mukanya, Delphine Mbonga-Mande, Lionel Baketana-Kinzonzi, Eddy Kinganda-Lusamaki, Daniel Mukadi-Bamuleka, Fabrice Mambu-Mbika, Placide Mbala-Kingebeni, Edith Nkwembe-Ngabana, Antoine Nkuba-Ndaye, Daniel Okitundu-Luwa, Steve Ahuka-Mundeke

**Affiliations:** 1Département de Virologie, Institut National de Recherche Biomédicale, Kinshasa 01204, Democratic Republic of the Congo; mathymatungala01@gmail.com (M.M.-P.); juniorbulapenge@gmail.com (J.B.-P.); merismatondo201@gmail.com (M.M.-K.); franckedidi@gmail.com (F.E.-A.); elisepukuta@gmail.com (E.P.-S.); doctapaul2010@gmail.com (P.T.-M.); stevsndombs@gmail.com (E.N.-M.); reginemande92@gmail.com (D.M.-M.); baketanalionel@gmail.com (L.B.-K.); eddylusamaki@gmail.com (E.K.-L.); drmukadi@gmail.com (D.M.-B.); mambumbika2@gmail.com (F.M.-M.); mbalaplacide@gmail.com (P.M.-K.); edithnkwembe1@gmail.com (E.N.-N.); amstev04@gmail.com (S.A.-M.); 2Service de Microbiologie, Département de Biologie Médicale, Cliniques Universitaires de Kinshasa, Université de Kinshasa, Kinshasa H8P3+7X3, Democratic Republic of the Congo; 3Département de Biologie Médicale, Université Protestante au Congo, Kinshasa 01212, Democratic Republic of the Congo; 4Division Provinciale de la Santé de Kinshasa, Kinshasa 01204, Democratic Republic of the Congo; samyesala@gmail.com (S.E.); trekat01@gmail.com (T.K.); 5Direction de Surveillance Epidémiologique de Kinshasa, Ministère de Santé, Hygiène et Prévention, Kinshasa 01204, Democratic Republic of the Congo; tutuyannick@gmail.com; 6Centre Neuro-Psycho-Pathologie, Kinshasa H8V3+CJ3, Democratic Republic of the Congo; daniel.okitundu@unikin.ac.cd

**Keywords:** paralysis, chikungunya, Kinshasa

## Abstract

Polio-associated paralysis is one of the diseases under national surveillance in the Democratic Republic of the Congo (DRC). Although it has become relatively rare due to control measures, non-polio paralysis cases are still reported and constitute a real problem, especially for etiological diagnosis, which is necessary for better management and response. From September 2022 to April 2023, we investigated acute flaccid paralysis (AFP) cases in Kinshasa following an alert from the Provincial Division of Health. All suspected cases and their close contacts were investigated and sampled. Among the 57 sampled patients, 21 (36.8%) were suspects, and 36 (63.2%) were contacts. We performed several etiological tests available in the laboratory, targeting viruses, including Poliovirus, Influenza virus, SARS-CoV-2, Enterovirus, and arboviruses. No virus material was detected, but the serological test (ELISA) detected antibodies against Chikungunya Virus, i.e., 47.4% (27/57) for IgM and 22.8% (13/57) for IgG. Among suspected cases, we detected 33.3% (7/21) with anti-Chikungunya IgM and 14.3% (3/21) of anti-Chikungunya IgG. These results highlight the importance of enhancing the epidemiological surveillance of Chikungunya.

## 1. Introduction

Acute flaccid paralysis (AFP) is a clinical syndrome characterized by the sudden onset of weakness of the lower motor neuron type, including weakness of the respiratory and pharyngeal muscles that progresses to maximum severity within several days to weeks [1]. It is a medical emergency because of its clinical progression, which can be either progressive, leading to irreversible paralysis or death, while some cases of paralysis go into spontaneous remission and are therefore reversible [2].

A number of etiologies have been proposed for paralysis, including non-infectious illnesses and infectious diseases such as viruses, which are the most common cause [3]. In Africa, particularly in sub-Saharan regions, this type of paralysis is commonly associated with enteroviruses (polioviruses and non-polio viruses) [4]. However, a few cases have been associated with respiratory viruses and arboviruses, such as Chikungunya virus (CHIKV) [5].

In the Democratic Republic of Congo (DRC), the AFP is under surveillance at the national level, and most of the cases are reported to be associated with poliovirus, which causes poliomyelitis. Although cases of poliovirus-associated paralysis have become relatively rare as a result of control measures such as adherence to hygiene measures and vaccination, cases of non-polio paralysis occur sporadically in certain regions of the country, but the etiology for most of them remains unexplained due to a number of factors, including the lack of a reliable, rapid, and accurate diagnostic test.

The diagnosis of paralysis is usually made based on epidemiological investigation using the case definition and a number of clinical and epidemiological characteristics of the paralysis [3]. Suspected cases are therefore notified as probable AFP cases at the national level to perform a test for a confirmation diagnostic and make better decisions for the specific treatment of the patients.

Recently, a series of AFP cases from the Kinshasa province were reported to the national surveillance system, while the city was not concerned by the recrudescence of poliomyelitis cases in recent years. Given the increase in the number of suspected AFP cases reported and above all, the unknown etiology of these manifestations, we considered it important to conduct epidemiological and biological investigations to identify the etiology of these cases to contribute to better management of AFP in Kinshasa.

## 2. Materials and Methods

### 2.1. Epidemiological and Clinical Investigations of the Suspected Cases of AFP

In early September 2022, the Ministry of Public Health, Hygiene, and Prevention (MoH) of the DRC received an alert on the occurrence of sporadic cases of paralysis in Kinshasa following the deaths of two boys aged 13 and 18 who presented with headaches, fever, and paralysis of the lower limbs. All of them were living in the Matete health zone (HZ). Epidemiological investigations by the Provincial Division of Health (PDH) of Kinshasa City documented the spread of the phenomenon within the same health zone and to the other health zones close to the Matete HZ.

From September 2022 to April 2023, a multidisciplinary team composed of epidemiologists, biologists, and clinicians from the Ministry of Health of the DRC was established to identify the etiology of this unknown phenomenon. We included any person presenting with functional lower limb impotence that could progress to paralysis, with or without fever, as an AFP suspect. We also included close neighbors of suspected AFP cases, as the unknown disease seemed to spread among people who shared the same area (household, church, and/or school). These neighbors were considered contacts of AFP cases.

AFP was diagnosed through a clinical examination performed by clinical neurologists. This examination focused on higher functions, cranial nerves, motor skills, sensitivity, balance, and coordination of movement, as well as the spine. Segmental strength, muscle tone, osteotendinous, abdominal-cutaneous, and plantar reflexes were also assessed. The Barrés test and Mingazzini test were performed.

### 2.2. Monitoring Contacts of Paralysis Cases

A contact was defined as any person in the immediate or direct environment of the suspected AFP case. They were monitored according to the following procedure: the alert (presence of paralysis case) was reported to the HZ (peripheral level), which passed it to the PDH (intermediate level) in Kinshasa, which informed the Epidemiological Surveillance Division (central level), responsible for investigating suspected case of diseases with epidemic potential. Once the alert was validated, we proceeded to identify and compile a list of contacts. These contacts were monitored clinically, with daily vital signs and clinical surveillance for motor and/or sensory deficits, and samples were taken for analysis.

### 2.3. Biological Investigations of the Cases

Our biological investigation consisted of searching for viruses implicated in the onset of paralysis according to our literature review [3,4,5]. These viruses were mainly polioviruses and non-polio enteroviruses A, B, C, and D using cell culture techniques, followed by respiratory viruses including Influenza and SARS-CoV-2, as well as human Coronaviruses (such as 229E-NL63-OC43 and HKU-1), and other respiratory viruses (such as RSV, HMPV, PIV1-2-3-4, EchoVirus, RhinoVirus, and AdenoVirus) using reverse transcription polymerase chain reaction. We also used RT-qPCR and ELISA serology for the detection of viral RNA and antibodies for arboviruses, respectively. Finally, we used metagenomic analysis to search for multiple viruses of medical interest (Figure 1). The full investigation approach and methods used are detailed in the Appendix A.

### 2.4. Statistical Analysis

We expressed qualitative variables in frequency or proportion, while quantitative variables were expressed as means with standard deviations. All processes, including data cleaning and analysis, were conducted using SPSS version 22.0. The geographical distribution of suspects and their contacts was analyzed using QGIS version 3.16.

### 2.5. Ethical Approval

Since this case was under a routine surveillance program conducted by the DRC’s national reference laboratory for outbreak investigation, the “Institut National de Recherche Biomédicale”, the Ethics Committee of the School of Public Health at the University of Kinshasa (DRC) was exempt from reviewing it. However, the School of Public Health’s Ethics Committee at the University of Kinshasa provided consent for the data to be used in scientific publications following the ethical approval number ESP/CE/173/2023.

## 3. Results

### 3.1. General Characteristics of Patients and Their Contacts

A total of 21 patients were notified as AFP cases at the national level. We also investigated their contacts (n = 36). Both suspected AFP cases and contacts were sampled to determine the AFP etiology. The overall age median was 20 (20–41), and the age median of the suspected AFP cases was 19 (12–41), while it was 21 (9–43.8) for the contacts of AFP cases (Table 1).

### 3.2. Clinical and Epidemiological Characteristics of Participants

We also investigated and collected the clinical symptoms of the AFP cases. Indeed, most suspect cases presented functional impotence of the lower limbs at 85.7% (18/21) and myalgia at 81% (17/21). We also reported non-specific symptoms in suspected AFP cases, such as cough, rhinorrhea, and headache in 28% (6/21) each (Table 1). According to the epidemiological investigation, we screened the geographic location of the suspected AFP patients. Indeed, all cases were scattered on both sides of the N’djili River, and the majority of them (16/21) came from the Matete and Limete health zones (Figure 1). In addition, we determined the location of the contacts and observed that they were found on both sides of the N’djili River, with a large number of contacts in the Matete (8/36) and Limete (6/36) health areas (Figure 2).

### 3.3. Investigation of the AFP Etiology

We performed Cell culture tests for poliovirus and non-polio enteroviruses and reverse transcription polymerase chain reaction (RT-qPCR) tests for respiratory viruses and arboviruses that were presumed to be the causes of the reported AFP. In fact, all tests were negative (no detection of the viruses or their antigens). We also performed a metagenomic analysis to search for over a thousand viruses of medical interest; this test was negative (Appendix A).

Finally, based on the epidemiological and clinical presentation of the suspected cases, we performed the Chikungunya Elisa test. The overall frequency of anti-IgM Chikungunya antibodies was 47.4% (27/57). Anti-IgM Chikungunya antibodies were present in 55.6% (20/36) of AFP contact cases and 33.3% (7/21) of AFP suspected cases (Table 1). The overall positivity for anti-IgG Chikungunya antibodies was 22.8% (13/57). We found 14.3% (3/21) of anti-IgG Chikungunya antibodies in AFP suspected cases and 27.8% (10/36) of positivity for IgG among AFP contact participants (Table 1).

## 4. Discussion

The present report aimed to describe the etiology of suspected AFP cases reported in Kinshasa between September 2022 and April 2023. The majority of AFP suspects presented functional impotence of the lower limbs, followed by myalgia. We also reported non-specific symptoms such as cough, headache, and rhinorrhea preceding the paralysis. These unspecific symptoms are also reported to be present in non-polio paralysis cases, such as during the onset period of Guillain–Barré syndrome [14].

The biological investigation for poliovirus detection showed that all AFP cases were negative for poliovirus and other enteroviruses, suggesting the involvement of non-polio etiology of AFP. Based on the proximity between most reported AFP cases, we suspected an infectious disease transmitted either by direct contact with the environment or close contact with the patient. We therefore extended the investigation to other viruses, such as respiratory viruses known to be present in Kinshasa, the capital of the DRC, among AFP suspected cases, and we included close contact samples in the screening. Indeed, Van Doorn PA et al. reported a number of paralysis occurring after influenza or gastrointestinal syndrome in 2008 [14].

Furthermore, we considered the location of the cases and their epidemiological history. All cases and their contacts were located around the N’djili River, particularly in a flood-prone area (Maziba district). This region was a hotspot for Chikungunya outbreaks in 1999 and 2015 [15,16]. This area is also known for its high mosquito population, which could justify or explain the strong suspicion of chikungunya [17]. Thus, we also extended the screening to arboviruses such as Chikungunya, dengue, and Zika.

The RT-qPCR tests for Influenza, SARS-CoV-2, and Chikungunya were negative. Given the clinical evolution of the cases, with the reversibility of AFP, we suspected AFP to be a late complication of the Chikungunya virus. We performed a Chikungunya serological analysis and detected anti-Chikungunya antibodies in the majority of suspects and their close neighbors. Indeed, Chikungunya has been reported as a cause of neurological manifestations. For example, in 2016, Talys J. Pinheiro et al. documented neurological manifestations, including paralysis, in 16% of 300 people infected with the Chikungunya virus in Polynesia and Brazil [18].

The non-detection of Chikungunya virus in our investigation study could be related to sample collection at an advanced stage of the disease, with a lower probability of detecting the viral material. Therefore, the serological approach provided better odds for diagnosis by detecting IgM and IgG antibodies in the blood. In addition, contacts could have presented paucisymptomatic and/or moderate forms of infection with the presence of IgM antibodies, while others would have been cured as IgG antibodies were detected. Hughes RAC et al. supported this hypothesis by showing in 2005 that in most cases, it is difficult to isolate and identify the agent responsible for non-polio paralysis, as the diagnosis is usually made at an advanced stage of infection or after infection and often only serological tests are positive [19,20]. We looked for and documented an etiology other than poliovirus in the context of recurrent paralysis cases in DRC with a lack of technical facilities, but the late investigations could justify the non-detection of viral ribonucleic acid (RNA). However, we did not investigate other biological disturbances (haematological, biochemical, etc.) or seroneutralization to confirm the functional activities of anti-Chikungunya antibodies detected. In addition, we cannot exclude the role of the SARS-CoV-2 vaccine in a portion of participants eligible for vaccination but who did not report their vaccination status.

## 5. Conclusions

We investigated suspected cases of AFP in Kinshasa from September 2022 to April 2023. All cases were non-polio paralysis, and we detected anti-Chikungunya antibodies in the majority of them, suggesting a neurological complication of the Chikungunya virus in this population. It is therefore important to enhance epidemiological surveillance, especially in high-risk areas, and conduct detailed laboratory testing. It would be important to report all cases at the early onset of non-specific symptoms, such as fever, at the surveillance system to enhance the probability of virus isolation and its molecular characterization.

## Figures and Tables

**Figure 1 pathogens-13-00198-f001:**
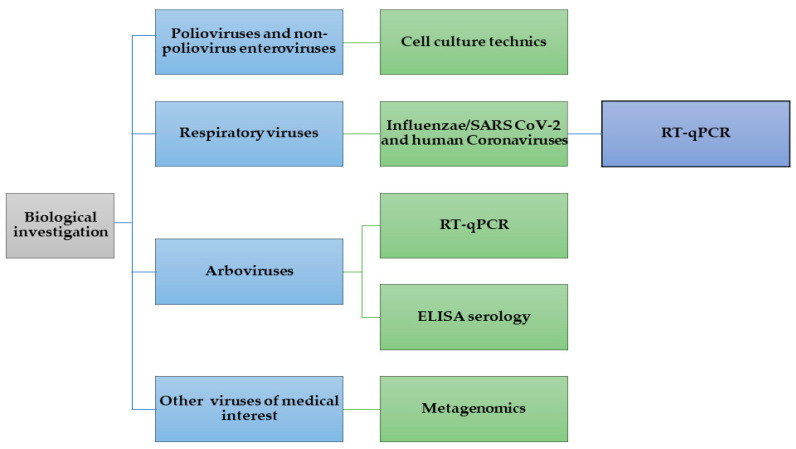
Diagram of biological investigations of AFP cases (designed following a non-standardized investigation approach, except for the detection of poliovirus) [6,7,8,9,10,11,12,13].

**Figure 2 pathogens-13-00198-f002:**
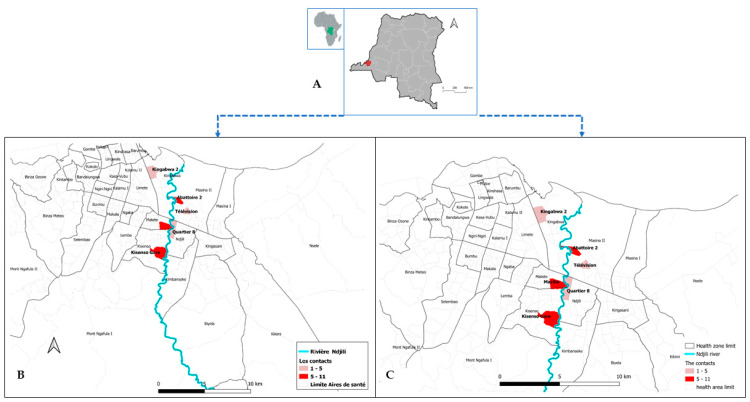
Map of Kinshasa in DRC (**A**) and Geographical distribution of suspects (**B**) and contacts (**C**) by Health Area.

**Table 1 pathogens-13-00198-t001:** General, clinical, and biological characteristics of AFP suspected cases and their close neighbors.

Variables	Total n = 57	Epidemiological Status	
		Contacts of AFP Cases n = 36	AFP Suspected Cases n = 21
Age (years), median (IQR)	20 (11–41)	21 (9–43.8)	19 (12–41)
Sex			
Female	34 (59.6%)	19 (52.8%)	15 (71.4%)
Male	23 (40.4%)	17 (47.2%)	6 (28.6%)
Symptoms			
Fever	5 (8.8%)	0 (0%)	5 (23.8%)
Cough	6 (10.5%)	0 (0%)	6 (28.6%)
Rhinorrhea	6 (10.5%)	0 (0%)	6 (28.6%)
Headache	6 (10.5%)	0 (0%)	6 (28.6%)
Myalgia	17 (29.8%)	0 (0%)	17 (81.0%)
Anorexia	5 (8.8%)	0 (0%)	5 (23.8%)
Nausea	2 (3.5%)	0 (0%)	2 (9.5%)
Functional impotence of lower limbs	18 (31.6%)	0 (0%)	18 (85.7%)
Functional impotence of lower and upper limbs	1 (1.8%)	0 (0%)	1 (4.8%)
Chikungunya Serology			
Anti-CHIKV-IgM			
Negative	30 (52.6%)	16 (44.4%)	14 (66.7%)
Positive	27 (47.4%)	20 (55.6%)	7 (33.3%)
Anti-CHIKV-IgG			
Negative	44 (77.2%)	26 (72.2%)	18 (85.7%)
Positive	13 (22.8%)	10 (27.8%)	3 (14.3%)

## Data Availability

Lab analysis and serological data are available on request to corresponding and senior authors.

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
