# Peer review of "Lower Limb Paralysis Associated with Chikungunya in Kinshasa, the Democratic Republic of the Congo: Survey Report"

_pathogens, 2024, doi:10.3390/pathogens13030198_

Round 1

Reviewer 1 Report

Comments and Suggestions for Authors

Review

In this short report, the authors investigated the potential causes of the increase in AFP cases in Kinshasa and their association with CHIKV. The report is an important alert, and it can be suitable for publication after the authors address the major points and minor corrections. Please, address the points listed below.

Abstract: 

The abstract is well-written, very informative, and objective.

Introduction:

Minor points:

1-    Line 41: “common cause[3].In”, add space. 

2-    Line 56: Plural or singular in: “to perform a confirmation tests”

3-    Line 56: Instead of “to perform a confirmation tests for a confirmation diagnostic”, a suggestion: “to perform a test for a confirmation diagnostic”.

Methods:

Line 92: Can the authors cite the references for: “of paralysis according to our literature review.”

Line 93 to 99: This section needs to specify that the methods are detailed in the supplementary materials, upfront. When I was reading it, I thought it was incomplete, lacking the details of the methods until I realized the information is all there in the supplement. The authors can also cite the used references for the applied methods. Please, cite them in this section when is convenient, for example, “using cell culture techniques…”, citation.

Figure 1 can also be improved (with references if the tests were standardized). 

Line 111 – Is there a number for the exemption protocol? This should be described in “the Ethics Committee of the School of Public Health of the University of Kinshasa  (DRC) was exempt from reviewing it.”

The authors used RT-qPCR (with TaqMan probes) but RT-PCR is mentioned (which is the conventional one). Please, fix this issue throughout the paper.

Results:

Lines 141-147 – How do the authors know the antibodies are not cross-reacting? Are the controls appropriate? 

Line 167 – Why did the authors just perform biological tests for poliovirus and enteroviruses, but not ELISA, as they did for CHIKV? This is critical for the interpretation of the brief report. The conclusions were based on IgG and IgM for CHIKV but no investigation of IgG and IgM was done for polio and enteroviruses. How can the authors discard this possibility?  Have the authors performed these tests? If not, the limitations of the study should be reported, and this issue should be addressed in the overall meaning of the study.

Lines 163 to 165: “We therefore, suspected the possible environmental origin of the disease, transmitted either by direct contact with the environment, by contact with infected people or by infected vector.”, I am very concerned with this sentence. Are the authors suggesting that Chikungunya is transmitted from person to person? Or environmentally transmitted? If this is the meaning, I do not think there is any evidence shown in this regard from the top of my knowledge. If the authors are trying to explain something else, this sentence needs to be written in another way. The present work does not show any data in this regard as well. 

Lines 203-209 – Please remove the extra spaces: “Supplementary Materials: The following supporting information can be downloaded at: 203 www.mdpi.com/xxx/s1, Figure S 1 : Diagram of biological investigations of the AFP cases ; Figure 204 S 2: Cell culture-based poliovirus diagnostic algorithm recommended by the WHO Global Polio 205 Laboratory Network (GPLN) ; Table S 1 : Possible results and notification of virus isolation results  in culture for an individual specimen ; Table S2: Primers and Probes for influenza (Protocol of the US Centers for Disease Control and Prevention) ; Table S3 : Primers and probes included in the Chikungunya RT-qPCR assay ; Table S4 : Outcomes of the biological investigations .”

Supplementary material: 

Line 125: “We extracted DNA at INRB using a Qiagen RNA Mini kit from blood samples and…”, Should it be “extracted RNA” here?

Reviewer 2 Report

Comments and Suggestions for Authors

Mathy et al. wrote an interesting article about the involvement of CHIKV in the functional impotence of the lower limbs by establishing the presence of CHIKV in those cases.  Very elegantly, the authors eliminated the involvement of other viruses by doing extensive analysis, which is commendable.

1.      It would be better for the reader’s perspective to include the assays performed to establish the functional impotence of lower limbs, as it was observed in 85.7% of cases.  

2.      How many days were the functional impotence of the lower limbs observed or persisted in those patients?

3. The reviewer would like to know if there is any prevalence of West Nile virus in the study area and if the authors thought about it.

4. Were any autoantibodies detected since the GBS-like symptoms were exhibited in these cases

5.      It would be better to provide the source of reagents for all ELISA assays performed for different flavi- and entero-viruses.  

6. The reviewer is wondering about the presence of immunoglobulins in 7 (IgM) and 3 (IgG) patients, as it suggests that maybe seven patients were in early infection (few days), and that leads to functional impotence of lower limbs, which is astronomical.

7.      Line number 69 would be enough to say about the health zone to confirm the locality of the patients.

8. Have the patients had SARS CoV-2 vaccination, particularly from Johnson and Johnson or AstraZeneca vaccine, as Mayoclinic suggests (https://www.mayoclinic.org/diseases-conditions/guillain-barre-syndrome/symptoms-causes/syc-20362793) vaccine can also elicit GBS like symptoms.  

Comments on the Quality of English Language

Language is readable and understandable. 

Round 2

Reviewer 1 Report

Comments and Suggestions for Authors

The authors addressed all my questions and concerns, and have improved the manuscript. I recommend the acceptance of the manuscript in its present form. 

Reviewer 2 Report

Comments and Suggestions for Authors

No comments to the authors, since they addressed the reviewer's concerns.